# A Review of the Role of Oral Microbiome in the Development, Detection, and Management of Head and Neck Squamous Cell Cancers

**DOI:** 10.3390/cancers14174116

**Published:** 2022-08-25

**Authors:** Kimberly M. Burcher, Jack T. Burcher, Logan Inscore, Chance H. Bloomer, Cristina M. Furdui, Mercedes Porosnicu

**Affiliations:** 1Wake Forest Baptist Medical Center, Winston-Salem, NC 27157, USA; 2Lake Erie College of Medicine, Bradenton, FL 34211, USA

**Keywords:** microbiome, squamous cell cancers of the head and neck, prognostic and predictive biomarkers, modulation of microbiome

## Abstract

**Simple Summary:**

Head and neck squamous cell cancers (HNSCC) are the seventh most common form of cancer in the United States. Though the role of the microbiome in the development of other diseases of the aerodigestive tract is well defined, the role of the microbiome in HNSCC is a developing subject. The notion of harnessing the microbiome for the prevention, detection, and treatment of HNSCC is an exciting new prospect in oncology. This manuscript discusses what is known about the healthy oral microbiome, the microbiome unique to premalignant lesions of the head and neck, the microbiome of HNSCC, and the microbiome as a modulator of immunity and malignancy. The manuscript also discusses clinical applications of the microbiome as they relate to HNSCC, including relationships between the microbiome and outcome data, and treatment toxicities. The aim of this review is to guide future research and clinical trials on the microbiome with the hopes of improving screening techniques, decreasing treatment toxicities, and improving survival for patients with HNSCC.

**Abstract:**

The role of the microbiome in the development and propagation of head and neck squamous cell cancer (HNSCC) is largely unknown and the surrounding knowledge lags behind what has been discovered related to the microbiome and other malignancies. In this review, the authors performed a structured analysis of the available literature from several databases. The authors discuss the merits and detriments of several studies discussing the microbiome of the structures of the aerodigestive system throughout the development of HNSCC, the role of the microbiome in the development of malignancies (generally and in HNSCC) and clinical applications of the microbiome in HNSCC. Further studies will be needed to adequately describe the relationship between HNSCC and the microbiome, and to push this relationship into a space where it is clinically relevant outside of a research environment.

## 1. Introduction

An estimated 2.2 million infection-attributable cancer cases were diagnosed worldwide in 2018 [1]. Historically, it has been accepted that the presence of a single organism may be the etiological agent of a given disease; however, recent insights highlight the methods by which dysbiosis, or the loss of balance within the commensal microbial community, creates an environment that may lead to cancer [2]. Various microorganisms (including bacteria, fungi, protozoa, and viruses) inhabit the human body, colonizing areas such as the mouth, nasal passages, intestinal tract, and skin [3,4]. These microbiota, in combination with their genomes and the surrounding environmental conditions, make up the microbiome [5].

The subject of the microbiome and its role throughout the progression of malignancy was first studied in colorectal cancers and focused on the intestinal microbiome. As data emerged, the relationship of the microbiome to the development, detection, and management of other malignancies has drawn the attention of researchers. The goal of such research is to harness the microbiome to prevent the development of cancer, achieve early detection, and increase tolerability and efficacy of cancer treatments.

The role of the microbiome in the development and course of head and neck squamous cell cancer (HNSCC) is of particular interest given the implication of a simultaneously symbiotic and predatory relationship between the affected parties (the typical and post-surgical oral structure, eubiotic microbiome, malignancy, carcinogenic microbiome, and treatment). Balance within the commensal microbial community enhances physiologic and mucosal immune functions, thus facilitating optimal bodily processes and overall health [6]. The interplay between the microbiome and the immune system also implies interplay between the microbiome and response to immunotherapy and other forms of treatment [4,7,8,9,10,11,12,13].

This paper reviews the available literature regarding the healthy microbiome of structures affected by HNSCC, changes in the microbiome that may serve as early markers of malignancy, abnormalities promoting malignancy, and changes to the microbiome that occur in response to treatment of head and neck cancers related to chemotherapy, radiotherapy, and surgery. We will also briefly review the role of the microbiome in response to immunotherapy. The authors of this manuscript postulate that the research found at the intersection of the microbiome and squamous cell head and neck cancers will allow for developments in the prevention and early detection of squamous cell head and neck cancers in addition to improvements in the safety, tolerability and potentially the efficacy of the anti-cancer treatments, thus reducing morbidity and mortality.

## 2. The Microbiome of Structures of the Aerodigestive System throughout the Development of HNSCC

### 2.1. The Healthy Microbiome of Structures Affected by HNSCC

HNSCC arises from various epithelial sites in the upper aerodigestive tract. A variety of microorganisms inhabits these areas. Each location along the aerodigestive tract differs from the next in terms of the quantity and variety of microbial species [14]. The oral cavity contains distinct microbial niches, including the lips, cheeks, tongue, teeth, gingival sulci, attached gingiva, hard palate, and soft palate [15,16]. There are several extensions adjacent to the oral cavity, such as the sinuses, nasal passages, trachea, lungs, tonsils, pharynx, larynx, esophagus, Eustachian tube, and middle ear. In understanding the relationship between these structures and the microbes that inhabit them, it is essential to first define some key components. A review by Marchesi and Ravel described the microbiome as an entire habitat, including microorganisms, their genomes, and surrounding environmental conditions. This same review defined microbiota as the variety of microorganisms in a particular environment, including bacteria, archaea, and lower eukaryotes [5]. The human oral microbiome, as the microbiome inhabiting and comprising the aforementioned spaces will be referred to throughout this manuscript, encompasses all the microbes, microorganisms, and structures present within the oral cavity and its adjacent extensions excluding anything beyond the distal esophagus [15].

The expected oral microbiota structure comprises more than 700 observed species, most of which have yet to be cultured [17]. The advent of next-generation sequencing has allowed the study of this microbiota with impressive resolution and throughput [18]. Symbionts are not only extracellular, but commonly external to the epithelium, and are often considered part of the host’s environment. Symbionts are thought to have co-evolved with the host and host immune system. They enhance the function of the host’s cells and organs [19]. The homeostatic state between the microbiota and the host’s immune system is called the eubiotic microbiota [20]. Healthy oral microbiota include an abundance of the phyla *Firmicutes*, *Proteobacteria*, *Bacteroidetes*, *Actinobacteria*, and *Fusobacteria* [21]. In addition, it is characterized by a dominant presence of the genera *Streptococcus* and *Haemophilus* in the buccal mucosa, *Actinomyces* in the supragingival plaque, and *Prevotella* in the subgingival plaque [17,22]. Other common, healthy constituents include *Gemelli*, *Lactobacillus*, *Moraxella*, *Campylobacter*, *Granulicatella*, and fungi (*Candida*, *Cladosporium*, *Aureobasidium*, and *Saccharomycetales*). Archaea (*Methanobrevibacter smithii*, *Methanobrevibacter oralis*, and *Methanosphaera stadtmanae*) contribute to the population living on the surface of teeth [23]. Less frequently, there may be a dominant presence of the genera *Veillonella* and *Neisseria* [24]. Different bacterial species are found in different locations within the aerodigestive tract related to the variety of adhesins, receptors, and metabolic requirements (including oxygen availability) which are present at any given level [25]. Microorganisms that inhabit one aerodigestive area disperse to adjacent epithelial surfaces of other sites [15]. Although only three genera (*Neisseria*, *Corynebacterium* and *Kingella*) are associated with a decreased risk for developing HNSCC, many contribute to the inhibition of maladaptive change [26,27,28]. The interplay between these organisms fosters the growth of other commensals that promote beneficial environmental and metabolic conditions for established microbiota and inhibit maladaptive change [23]. 

Though unique amongst individuals, the microbiome is significantly altered by the structure and function of the aerodigestive tract. Saliva is the primary mode of transportation of nutrients, peptides, and partially dissolved carbohydrates in forming oral biofilms. The quantity of saliva can cause changes in the microbiome [29]. Mucins, a glycoprotein component of mucus and saliva, contribute to the structure of the oral microbiome by inducing host-microbiome interactions such as the adhesion of microbiota to surfaces. In addition, mucins protect the epithelium from pathogens that seek to colonize it and provide a source of nourishment for growing commensal microbes [16]. 

In turn, the diversity and abundance of microbial species present within the aerodigestive tract significantly contribute to the structure of the expected oral environment. Symbionts complement the immune system through alterations of the host environment (pH, chemical structure, nutritional resources), assembling their environment within the host (including creating plaques and pseudomembranes), and through actions of the genetic material of the host and symbionts. For example, the oral microbiota that dwell within the oral cavity and oropharynx influence structure by regulating the overgrowth of indigenous pathobionts and the colonization of exogenous pathogens [6]. In addition, species of the *Streptococcus* genus, which are highly abundant in the oral cavity, influence the structure of the oral microbiome by their ability to alter the thickness of the oral mucosa, thus modulating infection risk [16]. 

The oral microbiome is unique among individuals, and a volume of research has been dedicated to studying evolutionary events that influence patterns of variation such as genetic drift, selection, migration, and recombination. The consequences of these events have well-established roles evidenced by antibiotic resistance, drug side effects, pathogenic biofilm formation, diet, sanitation, and health status. Diversity in oral microorganisms is also affected by the individual’s age, geographic location, habits, pH, micronutrients, and secretions. Little is known regarding how the microbiome changes over time and how the aforementioned components work together to result in the array of phenotypes and genotypes that comprise the human oral microbiome [3]. It is known, however, that a neonate receives its first microbiota from its mother. The route of delivery (vaginal versus cesarean section) is the first event to determine that individual’s microbiome. Over time, the microbiota and the neonate’s immune system co-evolve to reach homeostasis. The microbiota of younger individuals is affected most by the host’s genetics, environment, lifestyle, and dietary habits [30,31,32,33]. Conversely, in older adults, the microbiota is defined most by overall health and diet, with those with higher frailty or long-term stays in care facilities having less diverse bacterial residents [34]. Some research supports studying the oral microbiota of patients on a regional and cultural basis as such studies have revealed significant differences amongst such groups [25]. Figure 1 demonstrates factors which contribute to shaping an individual’s microbiome.

### 2.2. The Microbiome Unique to Pre-Malignant Lesions in Environments Developing HNSCC

An emerging facet of research involves delving into the possibility that oral microbes may serve as potential biomarkers for various malignancies and pre-malignant lesions [35]. Leukoplakia, erythroplakia, and erythroleukoplakia are conditions defined as pre-malignant stages of oral cancers or oral potentially malignant disorders (OPMD). Identifying unique organisms specific to each stage of carcinogenesis, from subclinical OPMD to early-stage HNSCC, may allow physicians to begin targeted treatment earlier in the disease process. Such initiatives would greatly reduce morbidity and mortality. As most available data provides only a correlative relationship without a causative mechanism, it has been suggested that the information be used for screening purposes rather than prevention. However, the identification of causative mechanisms may allow for intervention in the future and prevention of the development of OPMD altogether [24,36,37,38]. 

The signatures in the oral microbiome in patients with OPMD are unclear. Still, early developments have been made in determining what differentiates the OPMD microbiome from the healthy and HNSCC microbiomes. Lee and colleagues compared the oral microbiota of patients with OPMD to patients with known HNSCC. This study demonstrated that *Fusobacterium*, *Prevotella*, *Porphyromonas*, *Veillonella*, *Actinomyces*, *Clostridium*, *Haemophilus*, *Streptococcus* spp., and *Enterobacteriaceae* are commonly found in both OPMD and oral squamous cell carcinoma and that *Cloacibacillus*, *Gemmiger*, *Oscillospira*, and *Roseburia* were more abundant in the oral microbiota population of patients with OPMD and oral cancer than in the oral microbiota population of their healthy counterparts. Furthermore, this study found that an abundance of *Bacillus*, *Enterococcus*, *Parvimonas*, *Peptostreptococcus*, and *Slackia* significantly differed between OPMD and oral cancer samples. The study concluded that alteration in these microbial communities could be a predictive marker for the transition from OPMD to HNSCC [39]. Mok et al. described the microbiome of OPMD lesions as a stage in which healthy and cancer-associated bacterial communities overlap and suggested *M. micronuciformis* as the best target for screening for OPMD because it was detected only in the OPMD group [40,41]. Oral lichen planus, another OPMD, has been correlated to an increased population of *P.*
*melaninogenica*, *Porphyromonas*, and *Solobacterium* and a lower abundance of *Haemophilus*, *Corynebacterium*, *Cellulosimicrobium*, and *Campylobacter* in oral microbiota in comparison to healthy controls [42]. 

Identifying dysbiotic microbiota to serve as indicators of malignancy is complicated by the effect of metabolism and lifecycle of anatomically distant microbiota on areas local to the cancerous or precancerous lesion in question [43,44,45,46]. For example, potentially carcinogenic metabolites from gut microbiota may enter the portal circulation with nutrients and enter the general circulation where they can exert their full effects [46]. Other factors that may complicate these studies include the microbiome’s unique composition, which is dependent upon by geography, culture, and patient-specific factors (such as age, gender, diet, lesion location, smoking, and human papilloma virus (HPV) status). 

### 2.3. The Oral Microbiota Unique to HNSCC

When microbial diversity and abundance are diluted, the oral microbiome enters a state of structural imbalance [47]. Such dysbiosis yields a structure that favors immune dysregulation, placing an individual at risk for various diseases and systemic infections [6,47]. Although recent research has identified many changes in the oral microbiome unique to HNSCC, there is no consensus data at this time. The variety of results may well be explained by differences in sample type (oral wash, tumor tissue sample, oral swab, etc.), stage, treatment history, and population, as well as the factors discussed in the section above pertaining to OPMD (the effects of distant microbiota, geography, culture, and patient-specific factors). Several studies have shown some concordance, but no clear pattern has yet emerged. Specific genera, however, particularly *Fusobacterium*, *Capnocytophaga*, *Prevotella*, and *Peptostreptococcus*, are implicated by multiple studies, some of which are summarized below. 

Wang et al. utilized 16S ribosomal DNA to identify patterns that were significantly different in HNSCC tumor resection specimens compared to healthy specimens. Two hundred forty-two samples were studied, and it was found that tumor samples were depleted of *Actinomyces* and had a higher amount of *Parvimonas* when compared to healthy models. These changes were noted to be correlative with T stage, with higher T stage tumors producing samples that were more significantly different from their healthy counterparts than lower T stage tumors [48]. A similar study by Takahashi et al. studied the salivary samples of 60 Japanese patients with oral cancer and found *Peptostreptococcus*, *Fusobacterium*, *Alloprevotella*, and *Capnocytophaga* were more abundant. *Rothia* and *Haemophilus* were relatively less prevalent when comparing the microbiome of cancer patients to the control patients [49]. Another study demonstrated an overabundance of *Fusobacterium nucleatum* (the most significantly overrepresented species in the tumors in this study), *Pseudomonas aeruginosa*, and *Campylobacter* [50]. Several other studies have noted a shift in the microbiota in HNSCC. These changes included a higher proportion of *Streptococcus* spp., *Capnocytophaga gingivalis*, *Prevotella melaninogenica*, *Porphyromonas gingivalis*, *Peptostreptococcus*
*stomatis*, *Gemella* (*G. haemolysans*, *G. morbillorum*), *Johnsonella ignava*, *Lactobacillus*, and *Prevotella* spp. Still, in most cases, no causative role has been suggested, and little prognostic information has been rendered [23]. 

Data collection on the prognostic value of microbiota analysis in the context of HNSCC is in its infancy. A recently published study confirmed the aforementioned shifts in the microbiota of patients with oral cancer but also purported a relationship between depth of invasion and organisms present. The study demonstrated that abundances of *F.*
*nucleatum*, *Capnocytophaga sputigena*, *Porphyromonas endodontalis*, and *Gemella haemolysans* were significantly increased in patients with oral squamous cell compared with the controls and that the abundances of *P. endodontalis*, *Gemella morbillorum*, and *G. haemolysans* increased with increasing depth of invasion of malignancy suggesting a dose-related relationship. In contrast, the abundances of *P.*
*melaninogenica*, *Haemophilus parainfluenzae*, and *Neisseria flavescens* decreased with increasing depth of invasion suggesting a similar relationship [51]. A separate study that may provide prognostic value describes the abundance of *Schlegelella* and *Methyloversatilis* in HNSCC tumors as a marker of poor prognosis. The same study suggests findings of *Bacillus*, and *Lactobacillus* and *Sphingomonas* to be positive prognostic findings [52]. Finally, a study by Eun et al. found the microbiota of patients with metastatic oral squamous cell cancer (OSCC) to be high in *Prevotella*, *Stomatobaculum*, and *Bifidobacterium* and low in *Fusobacterium* [53]. Additional research is needed with regard to disease outcome and its relation to changes in the microbiome. 

In one study, an association between oral cancer and the microbiome was used to develop diagnostic tools for cancerous and precancerous lesions: Mager et al. correlated high salivary counts of *C.*
*gingivalis*, *P.*
*melaninogenica*, and *Streptococcus mitis* with active oral cavity SCC. Using this known correlation, they predicted HNSCC status with sensitivity and specificity greater than or equal to 80% in both matched and unmatched groups [54]. Conversely, Guerrero-Preston et al. recommended further studies to determine the role of *Lactobacillus gasseri/johnsonii*, *Lactobacillus vaginalis*, and *F.*
*nucleatum* in future screening tests for active HNSCC [55]. Lim et al. found a panel that included *Rothia*, *Haemophilus*, *Corynebacterium*, *Paludibacter*, *Porphyromonas*, *Oribacterium*, and *Capnocytophaga* could discriminate the oral rinse samples of HNSCC patients with oropharyngeal and oral cavity lesions from age-matched healthy counterparts [56]. 

A metanalysis by Yang et al. demonstrated that the genus *Lachnoanaerobaculum*, *Kingella*, and *Parvimonas* could differentiate an oral squamous cell cohort from a healthy counterpart. Further pathway analysis revealed that these loci were enriched for genes in regulation of oncogenic and angiogenic responses, implicating a genetic anchor to the oral microbiome in estimation of casual relationships with OSCC [57]. 

Other studies have sought to associate the microbiota in HNSCC with clinical and outcome data and found that a higher abundance of *F. nucleatum* is associated with lower tumor stage, lower recurrence rate, and improved disease-specific survival. In attempting to identify a mechanism to explain this outcome data, it was determined that the overrepresentation of *F. nucleatum* was associated with host gene promoter methylation, including hypermethylation of tumor suppressor genes LXN and SMARCA2, further supporting its potential role in prognostication [58]. Another study echoed this: Neuzillet et al. found that tumors that tested positive for *F. nucleatum* using 16S rRNA sequencing were more likely from older patients with less combined alcohol and tobacco use than those with oral cancer who tested negative. A nearly significant trend suggested that these patients had a lower rate of recurrence and more prolonged overall, relapse-free, and metastasis-free survival [59]. 

A growing amount of data suggests other, occasionally contradictory, causative roles between oral microbiota and HNSCC. Such studies have correlated oral microbiota populations with a high prevalence of *Fusobacterium*, *Prevotella*, and *Streptococcus* to laryngeal carcinoma. These studies relate to the idea that *Streptococcus* is an antagonist of *Fusobacterium* and *Prevotella* through its participation in nutrient transfer and metabolism [41]. HNSCC samples with an elevated abundance of *F. nucleatum* have also been shown to have an upregulated oncogenic Wnt/β-catenin pathway and downregulated immune system pathways [60]. This was further examined by Wu et al., who demonstrated that *Fusobacterium nucletum* plays a role in the development of some oral cancers by inhibiting β-catenin signaling and increasing the expression of TLR4 activation of the p21-activated kinase and cyclin D1 simultaneously contributing to increased inflammation and suppression of NKT cell activities thus promoting malignancy [61]. Mok et al. have also implicated *Prevotella* as a partner of *Fusobacterium* in fostering the development of malignancy in the throat by changing the microenvironment and biofilm formation [41]. Finally, *Candida albicans* has been found to facilitate the development of OSCC by inducing the production of matrix metalloproteinases, oncometabolites, and oncogenes in non-malignant cells [62].

Other studies have reported the multifactorial method by which *P.*
*gingivalis* may participate in the development of HNSCC. These studies state that *P.*
*gingivalis* can induce the expression of specific receptors, reduce T cell proliferation, facilitate immune evasion and promote proenzyme matrix metalloproteinase 9 expression, thus potentially influencing the progression and metastasis of HNSCC [23,63]. 

Some state that, though the exact mechanisms through which the microbiome changes in HNSCC are variable, the ultimate outcome is equivocal. Perera and colleagues demonstrated that though the microbiota between two comparable populations affected by HNSCC may differ compositionally, differences between the HNSCC groups were negligible at a functional level. They postulated that the microbiota in both HNSCC groups expressed proinflammatory attributes, including lipopolysaccharide biosynthesis and peptidases and though the processes differed amongst individual microbiota even within the species level, the end outcome was an “inflammatory bacteriome” which contributed to the growth of HNSCC [50]. Furthermore, though the studies describing the oral microbiota unique to HNSCC lay a necessary groundwork, it is essential to note that current methods of describing the microbiome merely create lists of organisms that do not accurately describe their location or distribution, including distribution into microbial communities such as biofilms [46]. A recent study attempts to characterize the microbiome of HNSCC based upon intra- and extraoral organisms but fails to reach a consensus with previously published work [14]. Furthermore, the complex interweaving of the many metabolic paths that originate from the oral microbiome has not been fully elucidated, and none can truly exist in vitro in any context resembling in vivo without the others. The microbiota of the healthy, premalignant and malignant microbiome of the head and neck can be reviewed in Table 1. 

## 3. The Role of the Microbiome in Sickness and Health

### 3.1. The Microbiome as a Modulator of Immunity

Microbiota have been shown to interact with the innate and adaptive immune systems, altering the host’s anatomy and physiology. Studies have shown that germ-free (GF) mice (mice without significant microbiota presence) have smaller and more rare mucus-producing goblet cells and thinner mucus layers. This lowers the efficacy of the barrier against invasion. GF mice also have been shown to have smaller mesenteric lymph nodes, poor lymphocyte binding, underdeveloped Peyer’s patches, and a lack of lymphoid follicles in the lamina propia [9]. 

One method by which microbiota may alter the immune system is via their surface structure. The molecules residing on the exposed surface of a microbe and the products of their metabolism are known as microbe-associated molecular patterns (MAMPs), which are detected by pattern recognition receptors (PRRs) such as Toll-like receptors (TLRs). The innate immune system depends on TLRs to mount an immune response against a microbe. The binding of a MAMP to a TLR activates a cascade of interleukins creating a pro-inflammatory response [20,64]. In their 2017 study, Albusleme and Moutsopoulos focused on the interaction between one such interleukin, IL-17, and the microbiome. IL-17 is well known to play a role in immune surveillance at mucosal sites, acting in protective and pathogenic roles via multiple signaling pathways. The core functions of IL-17 include maintaining barrier integrity, promoting host control over microbial growth, and creating the first line of defense to regulate the recruitment and proliferation of neutrophils [65,66]. The microbiome’s influence on innate immunity is complex, using several synergistic mechanisms, including cell signaling cascades, modulation of tissue-specific mediators, hormonal regulation, hematopoiesis, and metabolic products. The results of this interaction include the fortification of host barriers, recruitment and maturation of immune cells, cell growth/death homeostasis, and other epigenetic modifications that result in a functional immune system that can detect and fight both infection and malignancy [67]. The effect of the microbiome on the innate immune system is shown in Figure 2.

Ultimately, the job of the innate immune system is to triage threats and activate the adaptive immune system only when appropriate pathogens are present; however, microbiota are also able to activate the adaptive immune system both directly [68,69] and through products of their metabolism [70]. The activation of the adaptive immune system leads to a durable change in the microbiome that can be a benefit or detractor. 

Chervonsky broadly described the complicated effects of symbionts on host health by recognizing the reciprocal interactions between commensals and the immune system, commensals and metabolism, and the interplay between the immune system and metabolism. He called this model the “Commenselocentric View of the Homeostatic Maintenance of Host Health” [71]. The homeostatic level of inflammation eventually reached by the cumulative impact of the interactions between the players that comprise the microbiome is known as immune tone. Immune tone is the readiness at which an immune response is potentiated [71]. This is briefly summarized by Figure 3. States of increased immune tone have been related to autoimmune disease, and cellular damage secondary to a proinflammatory state [71], whereas decreased immune tone is related to increased risk of infection and malignancy. The local and distant effects of the microbiota on the immune system are crucial to immune function, and it is essential to recognize that microbiota in the gut may affect the immune function in areas affected by HNSCC [9]. 

### 3.2. The Microbiome as a Modulator of Malignancy 

A growing body of literature explores how microbiota enhance the immunosurveillance and immune cell infiltration of malignant and pre-malignant lesions. This is particularly important due to immunotherapy’s curiosity and success (discussed later in this manuscript) [72]. Though most literature focuses on how microbiota induce inflammation and cause cancer, the methods by which the microbiome alters immune tone and immunosurveillance, leading to the clearance of malignant and pre-malignant cells, is a fascinating topic still in development [73]. Correlative studies have contributed to the development of the “hygiene hypothesis”, which states that reduced exposure to infections may lead to an increased risk of allergies and some autoimmune diseases (specifically type 1 diabetes and systemic lupus erythematosus). The hygiene hypothesis has been adapted and the findings supporting it extrapolated to create what is now known as the “cancer hygiene hypothesis.” This hypothesis suggests that the increase in the incidence of some cancers may be related to lifestyle changes. It postulates that the increased intake of sterilized and processed food has led to decreased exposure to certain microbial species thereby weakening the immune system and allowing cancer to reign [73]. 

Given the close interplay between the immune system and the microbiome, research increasingly focuses on the association between the composition and function of the microbiome and cancer [38]. Broadly, research has proposed the microbiome as a critical influence in cancer prevention and development due to the microbiota’s effect on immunosurveillance and immune tone and how it metabolizes chemical carcinogens [73,74]. Most theory focuses on mechanisms in which the microbiome of the gut modulates malignancy (both within the gastrointestinal system and distally) with the help of the unique architecture of the gut’s immune system. This is a prime focus for research as most of an individual’s microbiome and a large proportion of the immune cells reside in the gut. The gut contains as many bacterial cells as the combined number of cells in a human’s entire body [75]. Travel of immune cells and microbial metabolites originating from within the gut influence immune tone and cancer development in distal sites [46]. The microbiome itself and the downstream effects of the microbiome on immune tone have the potential to serve as either a protector or an aggressor, with its exact cumulative effects complicated by its many constituents and confounders. 

Though the idea of the microbiome as a protective agent is a well-known phenomenon in the world of infectious disease, this concept is more nebulous in oncology. A growing amount of literature, however, now supports this claim. For example, probiotic bacteria are widely accepted as protective agents against colorectal cancers due to their ability to modulate inflammation and phagocytosis. Many cohort studies illustrate the positive effects of dairy intake on reduced colorectal cancer risk [76]. Moreover, it has been demonstrated that the human oral microbiome participates in commensalism, which confers a level of protection from pancreatic cancer to a healthy individual [17]. Research has shown that changes in the microbiota in oral cancers can influence the activity and population of immune cells within a tumor [58,59].

However, the microbiome may foster malignancy in a pathogenic, dysbiotic state. In contrast to the eubiotic microbiome, the dysbiotic microbiome has deleterious effects on its surroundings and the host immune system. Research has identified several risk factors for dysbiosis, including exposure to antibiotics early in life, frequent use of antibiotics, diets high in fat and protein and low in fiber, and unhealthy habits like increased alcohol use or smoking [77]. Dysbiotic changes in the oral microbiome have been correlated to cancers at distant sites. For example, periodontal disease, characterized by tooth loss and inflammation within the oral cavity, has been recognized as a risk factor for pancreatic cancer [17,24,38,78]. In the case of gastric cancers, Hu et al. observed the tongue coatings of patients with gastric cancer versus those of healthy controls and found that in comparison with their healthy counterparts, the tongues of patients with gastric cancer exhibit a considerably thicker coating, which is correlated to a decline in microbial diversity. The investigators theorized that this deviation from the healthy microbiome allows carcinogenic species to emerge, thrive, and influence inflammatory responses, ultimately fostering carcinogenesis [24]. 

However, the mere prevalence of a species in cases of HNSCC cannot prove a causative relationship, though some studies have elucidated such a relationship and have built a case implying causation. For example, Jin et al. studied the changes to the microbiota inhabiting mouse models induced with lung cancer. The study found that these mice possessed a microbiome with lower species diversity compared to their healthy counterparts. Interestingly, transfer of the cancer-associated microbiota from experimental mice into GF mice lead to increased cancer incidence in these mice. Furthermore, the transfer of microbiota from mice with advanced lung cancer to those with early-stage lung cancer accelerated the growth rate of the malignancy in the early-stage mice. Both of these findings suggest the ability of the microbiome to augment carcinogenesis [79]. In a separate study, Frank et al. found that depletion of the disease-associated microbiota in mice delayed carcinogenesis while microbiota transfer from cancer-ridden mice quickened the process [80]. Other studies have suggested a bidirectional relationship between malignancy and the microbiota [20,61,81,82,83]. 

In 2015, Garrett derived three categories to characterize how the microbiome can influence carcinogenesis by either heightening or reducing an individual’s risk for developing cancer, which are as follows: (I) the microbiome alterations to the host cell proliferation/cell death balance, (II) influence on immune function, and (III) changes in the metabolism of host-produced factors, ingested nourishment, and pharmaceuticals [35]. Though this is amongst the most widely accepted and detailed theories for the carcinogenic and anti-carcinogenic effects of the microbiome, it is by no means all-inclusive. Other prevalent theories include adaptations made to Ewald’s barrier theory [19]. The microbiome’s influence on malignancy is therefore discussed below in a mechanism-based approach. 

Microbiota can alter the delicate balance between host cell proliferation and death by the microbiome, specifically altering the genomic stability and shifting the balance toward cell proliferation and away from cell death. This is the well-known mechanism used by the HPV oncovirus for the causation of HNSCC. Other examples in HNSCC include *P.*
*gingivalis*, which is known to have an antiapoptotic effect through its modulation of several pathways, including intrinsic mitochondrial apoptosis pathways, acceleration of the cell cycle and reduction in p53 levels, epithelial cell upregulation, and promotion of cell growth, neovascularization, metastasis, and secretion of inflammatory cytokines. Furthermore, *P. gingivalis* is known to secrete the anti-apoptotic enzyme called nucleoside diphosphate kinase [84,85]. *F.*
*nucleatum*, a microbial species known to be more abundant in HNSCC samples compared to samples from healthy counterparts, also alters the proliferation/death balance by activating kinases that ultimately lead to the proliferation of oral epithelial cells [86].

Microbiota are also able to alter the immune tone. In a functioning symbiotic relationship, the microbiota are shielded from the immune system by a mucosal barrier and vice versa. Breaks in the mucosa allow for the invasion of the surface by the immune system, the tissue with the microbiome. Consequently, the microbiome and the immune system find themselves in an environment in which they have not coevolved. The resultant microenvironment leads to either a pro-inflammatory or immunosuppressive host response paving the way to carcinogenesis through DNA damage (resulting in haphazard mutation of host cells) and impaired antitumor response, respectively. There is evidence that both higher and lower grade levels of inflammation are related to increased reactive oxygen and nitrogen species, cytokines, and chemokines that assist in the evolution of malignancy as support for this mechanism [35]. Another example is the activation of nuclear factor κB (NF-κB). An essential element of bacterial carcinogenesis, NF-κB is activated following a series of events trigged by bacterial endotoxins. Activation of NF-κB induces inflammatory-associated cytokine production and is widely considered carcinogenic [87,88]. Activation of tumor necrosis factor α (TNF-α) and interleukins also plays an essential role in the carcinogenic actions of some microbiota [74]. Other studies have pointed to mechanisms through which specific bacteria, such as *P.*
*gingivalis* and Fusobacteria, upregulate the immune response by either their structural components (ex: flagella) or their ability to alter host DNA transcription [89,90,91,92,93,94]. Smoking has been shown to lead to unstable microbial colonization and increased risk of bacterial infection through alterations in the innate and adaptive immune response and this has been extensively implicated as a potential mechanism causing HNSCC [95,96]. Additionally, some microbiota may induce inflammation by triggering a reaction of the innate immune system after being bound by PRRs [2,97]. Ultimately, the innate immune system participates in cycles which activate the adaptive immune system, thereby propagating an inflammatory, pro-malignant microenvironment via a targeted approach [98,99,100,101]. This self-serving cycle creates a proinflammatory state rich in molecules that serve as electron acceptors allowing microbiota that have evolved to thrive in these settings to gain prevalence. These bacteria then work to further this inflammation to give themselves a survival advantage and outcompete normal flora [102,103,104,105,106,107,108,109]. Conversely, other dysbiotic microbiota may promote the survival of cancerous cells by downregulating the host immune response, thus allowing tumor cells to go undetected [84,110,111,112].

Another widely accepted mechanism suggests that chronic inflammation is a method by which oral microbiota influence carcinogenesis without capitalizing on imperfections in the underlying mucosa. Dysbiosis in cancer sites is characterized by a notable decline in the diversity and richness of commensal bacteria. Disturbances such as this have two proposed effects: the rise of carcinogenic species and the induction of inflammation. When cancer causes a loss of commensal species, there may be a breach in the protection supplied by these species. Furthermore, competition for space and resources may be lost between commensal and carcinogenic species. This competition may serve to keep carcinogenic species at bay, therefore, carcinogenic species may thrive. Chen et al. suggest dysbiosis may induce and contribute to inflammation, which indirectly contributes to carcinogenesis [38]. There are various ways dysbiosis may result in carcinogenic inflammation. For example, dysbiosis promotes pathogenic microbes that possess the capability to penetrate, attack, and colonize host epithelial cells, thus inducing inflammation [35]. In addition, dysbiosis and the expansion of pathogenic microbes may cause inflammation upon the host’s innate immune system’s recognition of imbalance within the commensal community and the presence of invading species [4].

Bacterial metabolism also contributes to carcinogenesis through the production of metabolites or co-metabolites. Alterations in the microbiome may promote toxic microbes and byproducts, which may lead to abnormal metabolism in host cells and shifts in signaling pathways, ultimately altering cell signaling, proliferation, death, and immune response [35]. Such examples are gallic acid (a metabolite produced by the microbiome in the gut that could help to incite intestinal cancers in APC mutant mice with mutations in Tp53 through the WNT pathway) [113], and butyrate (a short chain fatty acid produced by bacteria in the gut) which can cause growth inhibition and apoptosis in HNSCC cells [114,115] and regulate galectin-1 content in HNSCC cells (thus influencing morphologic changes, cellular interactions, and differentiation) [116]. Garrett’s third mechanism can also be examined in the effects of alcohol, a known risk factor for HNSCC, on the microbiome. In the pathway of alcohol metabolism, microbiota (specifically *Streptococcus salivarius*, *alpha-hemolytic Streptococci*, *Corynebacterium*, and *Stomatococcus*) assist in the conversion of ethanol to acetaldehyde, a mutagen also found in tobacco smoke that initiates the formation of cancer within the head and neck mucosa through overstimulation of cellular regeneration and mucosal inflammation [37,38,117]. It has been theorized that repeated exposure to tobacco smoke leads to the natural selection of microbiota capable of a high rate of acetaldehyde metabolism, thus conferring tolerance to acetaldehyde. It is also known that smokers harbor changes in their oral flora that result in the production of more acetaldehyde from ethanol. This confers that the saliva of smokers has an increased potential for harboring more microbial flora capable of producing a known carcinogen. Thus, the oral bacterial flora may synergize with the primary risk factors such as alcohol abuse and smoking in the oral cancer pathogenesis [118]. Similarly, the microbiota of smokeless tobacco users with confirmed OSCC was found to be significantly different (favoring nitrosamine-forming bacteria, including *Staphylococcus*, *Fusobacterium*, and *Campylobacter*) than those without oral cavity SCC, regardless of smokeless tobacco use [119]. Others have studied the relationship between the oral microbiome and nitric oxide homeostasis as nitric oxide is considered a “double-edged sword” as some aspects of NO signaling demonstrate effects that lead to tumor growth, and other aspects having an anti-tumor effect [120]. Finally, some microbiota (*P.*
*gingivalis*, *P. intermedia*, *Actinomycetemcomitans*, and *F. nucleatum*) produce volatile sulfur compounds, which induce chronic inflammation, cell proliferation, migration, invasion, and tumor angiogenesis [85].

Rastogi et al. postulate that microbiota act in cohort with one another and that groups that find a neoplastic environment suitable for their survival often simultaneously use many mechanisms in tandem with one another to perpetuate the neoplasm. They coined these cohorts “oncogenic bacterial clades” and cited the use of cyclomodulins to disturb cellular polarity, cytoskeletal structure, and the balance of cell proliferation and cell death in addition to upregulation of oncogenes and generation of reactive oxygen species (ROS) as methods by which a clade reinforces the changes in a neoplastic environment that have given them an evolutionary advantage. Though the mechanisms recognized by Rastogi et al. are similar to those discussed above, the concept of a bacterial clade is a novel descriptor depicting the complexities of the microbial community highlighting interactions between microbiota and exposures [20]. 

Others have pointed to the adverse effects of genotoxins on eukaryotic host cells. In contrast to metabolites and co-metabolites generated incidentally, described in method three of the Garrett theory above, genotoxins are produced by bacteria with the sole purpose of causing irreparable DNA damage giving the creator a competitive advantage to survive. This DNA damage influences tumorigenesis [46]. Examples of such genotoxins include Colibactin (a product of *Escherichia*
*coli*) [121] and cytolethal distending toxin (a product of *Proteobacteria*) [122]. Neither of the latter two modalities have specifically been described in HNSCC.

Finally, some bacteria and viruses can insert portions of their genome into the DNA of eukaryotic host cells. In a study of 100 tumor tissue samples of patients with HNSCC of the oral cavity, genomic elements of *Mycobacterium*, *Sphingomonas*, *Campylobacter*, *Aeromonas*, *Bordetella*, *E.*
*coli*, HPV16, and JC Polyomavirus were found to be integrated into the chromosomes of oral cavity SCC cells. Some of these insertion points disrupted the host DNA in genes known to be related to the regulation of malignancy. For example, *Mycobacterium* was found to have insertion sites in the exonic portions of the tumor suppressor ADAMTSL1 and *Aeromonas* in the exon of the RASSF5 (a member of the Ras association domain family that functions as a tumor suppressor). Other insertion sites disrupted elements of cell cycle regulation and other tumor suppressor genes. Fungal oral cavity SCC flora were also detected at 125 insertion sites in the intergenic, intronic, upstream or downstream of genes or ncRNA but not exonic regions in the human chromosomes. These fungal elements were also able to disrupt the intronic regions of tumor suppressor genes. It is unclear at this time how these insertions may change the expression and products of these genes [123].

Others have cited the mechanisms microbiota use to inhibit the development of malignancy [19,124]. Ewald et al. have considered methods by which symbionts may aid the immune system in anti-cancer response. These mechanisms include supporting the responsiveness of immune checkpoints by postulating that the balance maintained by immune checkpoints is derived from the natural selection of microbiome components: an overactive immune response through inhibition of checkpoints would lower an organism’s ability to thrive through the destruction of healthy functional cells. In contrast, a suppressed immune response through activated checkpoints would leave the organism susceptible to infection and malignancy. The authors maintain that microbiota help regulate this responsiveness. Furthermore, they argue that current evidence indicates eubiotic microbiota exercise their anti-cancer effects through the enhancement of immune surveillance, inhibition of oncogenic viruses, and inhibition of pathogenic microbiota invasion rather than through the activation of barriers [19]. Overall, the theory states that the capability of microbiota to cause or prevent cancer relies on the microbiota’s evolutionary path to break down or maintain the barriers described by the theory or their ability to affect the existing host protective systems [125]. 

Overall, the case for a clear balance between benefit and harm of microbiota is complicated and must be considered on a malignancy by malignancy, organism by organism basis if not patient by patient (to account for changes from patient, and tumor-derived factors such as stage, geography, culture, smoking status, HPV status, etc.). Research and theory have proposed several feasible, generalizable mechanisms despite the limitations. 

## 4. The Microbiome and Treatment and Prognosis in HNSCC

### 4.1. Changes to the Oral Microbiome as a Response to Treatment of HNSCC

Treatment of HNSCC is complex and often includes combinations of surgery, radiotherapy, chemotherapy, and immunotherapy as determined by the stage and location of the primary lesion as well as specialist judgement and patient goals. Each of these treatment modalities may impact the oral microbiome as summarized in Table 1. 

Though very little research was identified regarding postoperative changes in the oral microbiota of adults with HNSCC, there is information regarding colorectal cancer patients who underwent surgical intervention, which has revealed that the microbiome following surgical intervention is unique and without resemblance to the population of patients studied with colorectal cancer or the healthy population. Changes in the gut microbiome following surgical intervention demonstrated lower diversity and fractured frameworks in pathways in which members of the gut microbiome react with one another [126]. A limited number of studies related to the oral microbiome of postoperative HNSCC patients. These suggested that postsurgical changes in oral microbiota of patients with HNSCC include overrepresentation of *Haemophilus*, *Neisseria*, *Aggregatibacter*, and *Leptotrichia* in patients’ saliva with HNSCC after surgery [127]. These changes have been shown to revert with the eradication of cancer which results in an increased prevalence of commensal bacteria (and no significant difference from pre-operative samples) three months following curative resection [128].

The oral microbiota, including the microbiota within biofilms, are known to be significantly changed during radiation therapy. These changes differ from those associated with the oral microbiota of patients who have completed radiation therapy and the pre-radiation population. Specifically, subjects receiving radiation had supragingival biofilms with a relative reduction in the abundance of Gram-negative obligate anaerobes and an increased abundance of *Streptococcus*
*mutans* [129]. Studies have shown that the relationship between the oral microbiome and radiation dose is inversely proportional; as the radiation dose increases, the diversity of the oral microbiome decreases. Likewise, these studies also demonstrated that the diversity of the oral microbiome increased over time following the cessation of radiotherapy [130]. It has been shown that radio-chemotherapy weakens the mouth’s defense mechanisms and may be partially responsible for the marked shifts in the oral microbiota [6]. Specifically, radiation therapy alters antibacterial properties, fostering detrimental oral microbiome alterations. Radiotherapy leads to acidification of the oral environment favoring the propagation of acidogenic and cariogenic species such as *Actinomyces*, *Lactobacillus*, and *S.*
*mutans*. This acidification also decreases the presence of *Neisseria*, *Fusobacterium*, and *Streptococcus sanguinis*. These changes contribute to the increased risk of dental caries in patients receiving therapeutic doses of radiation to the head, neck, and oral cavity [6,22]. Furthermore, the abnormal oral microenvironment facilitated by radiotherapy facilitates the growth of populations of opportunistic pathogens such as *Candida* species, enteric rods, and *Staphylococci* leading to an increased risk of infection [131]. 

Chemotherapy has been shown to decrease diversity within the microbiome and increase the risk of infection for those receiving treatment for other malignancies; however, the effects of these changes in HNSCC remain unclear [132,133,134,135]. Generally, patients receiving chemotherapy experience a difference in the oral microbiome; specifically, the relatively high population of oral *streptococci* decreases, and the shift favors an abundance of Gram-negative anaerobic flora, which are more pathogenic and implicated in inflammatory processes and ulceration [16]. Chemotherapy has also been shown to result in dysbiosis, which depletes genera responsible for regulating the enterosalivary nitrate–nitrite–nitric oxide pathway leading to measurable chemical changes (including catalyst concentration) within the mouth. This change could long outlast a patient’s treatment and may alter their risk of recurrence [136]. Patients with HPV-positive oropharyngeal cancers treated with combination chemoradiation therapy have been shown to significantly reduce the species richness and increase the relative abundance of gut-associated taxa in oropharyngeal swabs without any effect on the gut microbiota [137].

Immunotherapy with PD-1 inhibitors is now considered the standard of care in platinum-refractory HNSCC. Rather than exploring the effects of immunotherapy on the microbiome, most research has focused on the ways the microbiome affects immunotherapy in a concerted effort to identify methods to improve treatment response. This is discussed in detail below. Generally, however, the authors of this review postulate that inflammation evoked on any system secondary to a heightened immune response may affect the microbiome’s diversity and otherwise alter the metabolism and communication between its constituents. 

The effect of HNSCC treatments on oral microbiota are summarized in Table 2. 

### 4.2. The Microbiome in HNSCC Outcomes 

Traditionally, HPV and Epstein–Barr virus (EBV) have been microbial predictors of outcome in patients with HNSCC (with HPV tumor positivity being a predictor of better outcome and tumors with high EBV deoxyribonucleic acid (DNA) levels correlated with worse prognosis) [138,139]. However, there is some data that suggest that levels of certain bacteria can also affect treatment outcomes as summarized in Table 2. As previously stated, newer data has demonstrated a nearly significant trend in which patients with tumors that tested positive for *F. nucleatum* based on 16S rRNA sequencing had a lower recurrence rate and more prolonged overall survival, relapse-free, and metastasis-free survival [59]. 

The microbiome has long been known to dictate response to chemotherapy in other malignancies, conferring chemoresistance in colorectal cancers, [140] and modulating the response of chronic lymphocytic leukemia to cyclophosphamide [141,142]. Iida et al. studied how commensal bacteria influenced the response of lymphoma, colorectal cancer, and melanoma to oxaplatin. The study demonstrated that GF mice could not produce a sufficient amount of ROS and cytotoxicity in response to platinum-based chemotherapy, thus severely curtailing tumor response [143]. Platinum-based chemotherapies, particularly cisplatin, are a cornerstone in treating HNSCC. Cisplatin sensitivity has also been shown to be enhanced in cellular models by treatment with sodium butyrate, a short chain fatty acid produced in the gut [144]. Conversely, oral inoculation with *P.*
*gingivalis*, a bacteria frequently found to be abundant in HNSCC and periodontal disease, was found to induce chemoresistance in mice with HNSCC. These mice also had higher serum levels of IL-6, an inflammatory interleukin. Treatment of these mice with anti-inflammatory medication increased the chemosensitivity in the study population, supporting the theory that pathogen-induced inflammation in chemoresistance in HNSCC [145,146].

Response to radiotherapy is at least partially postulated to be affected by dysbiosis through changes in the metabolism of butyrate and other short chain fatty acids (SCFA) that result from microbial metabolism in the gut [115]. A study on mouse models in 2019 further characterized this claim. In the study, mouse models with melanoma or lung/cervical cancer were pre-treated with either oral vancomycin (providing adequate antibiotic coverage of gut microbes producing butyrate) or neomycin/metronidazole (which should not affect butyrate-producing organisms) before receiving radiotherapy. The mice who received vancomycin had a better response to radiotherapy, which was abrogated by the reintroduction of dietary butyrate. No change in response to radiotherapy was noted in the group pre-treated with neomycin/metronidazole [147]. Given the widespread effects of butyrate on many systems, it is likely these effects can be extrapolated and applied to HNSCC, but no trials have confirmed this.

Given the role of the gut microbiota in modulating immune response as detailed above, there is very little surprise that gut microbiota have been related to differences in treatment response to immunotherapeutic agents in various malignancies. Recent research has demonstrated that a healthy population of symbionts is critical to the response to immunotherapies. Treatment with antibiotics, which eliminate or otherwise alter the normal microbiota and microbial relationships, has been shown to affect response to immunotherapy adversely [143,148,149,150]. This theory suggests healthy intestinal microbiota synergize with the host’s immune system’s response to immunotherapy. In contrast, dysbiotic microbiota affect immune tone by either failing to provide immune stimulation needed to prime the innate immune system or through the presence of overabundant immunosuppressive methods that dysbiotic species have created to promote their survival [9,10,12,13,151]. However, despite promising results in other types of cancer, a recent phase 3 trial studied differences in the oral microbiome in patients with platinum-refractory HNSCC receiving nivolumab and found no statistically significant association between microbial population and clinical response [152].

Conversely, one source found that the prevalence of specific microbiota may be able to predict HNSCC response to immunotherapy. This source found that patients with HPV-positive, stage 3 oropharyngeal disease were significantly more likely to have a decreased abundance of *Akkermansia* in their stool samples [153]. This source cited observations that the presence of *Akkermansia* in stool samples can be related to the treatment outcomes of distant malignancies with PD-1 blockade [149,154,155]. Zheng et al. found that both mice and humans whose OSCC tumors contained an abundance of *Peptostreptococcus* had better long-term survival. Additionally, manipulating the microbiota to select for *Peptostreptococcus* in mice treated with PD-1 blockade therapy augmented results [156]. Alternatively, another study found that treatment with antibiotics (and resultant disruption of the microbiome) in the thirty days preceding the initiation of immunotherapy for the treatment of recurrent or metastatic HNSCC was significantly associated with decreased survival when compared to patients who were not treated with antibiotics in the thirty days before immunotherapy initiation [157]. 

The effect of oral microbiota on treatment outcomes in HNSCC are summarized in Table 3.

### 4.3. The Microbiome in HNSCC Treatment Toxicities 

Oral mucositis (OM) is a cancer treatment toxicity of radiation and chemotherapies characterized by erythema, edema, the formation of a pseudomembrane, mucosal shedding, ulceration, and bleeding traditionally thought to be initiated by oxidative stress with resultant cell death. The activation of the innate immune response is known to induce OM. The oral microbiome influences the innate immune response. For this reason, it is plausible to infer that the oral microbiome may affect the OM pathway, intensifying or lessening the symptoms and duration of OM as advantageous and disadvantageous changes are made to the microbiome. Oral microbes are accepted strictly as a modulating factor of mucositis, not a causative factor, but several studies have looked at the effects of such modulation, as summarized in Table 3. A cohort study in patients with locoregional HNSCC demonstrated that changes in oral microbiota genera over the course of treatment (particularly those favoring an overabundance of *Prevotella*, *Fusobacterium*, and *Streptococcus*, *Megasphaera*, and *Cardiobacterium*) were associated with the onset of OM [158]. Zhu et al. demonstrated that as OM advances to peak severity, the presence of Gram-negative bacteria increases [6]. As members of the *Proteobacteria* phyla, many of these Gram-negative bacteria are known to be opportunistic pathogens associated with mucosal ulceration. The findings of this study demonstrated that the abundance of Gram-negative bacteria has an aggravating effect, promoting mucosal inflammation; however, such influence cannot imply that oral microbes are an etiological factor [6]. Another study demonstrated that the severity of OM is associated with xerostomia and the presence of members of the family *Enterobacteriaceae* and genus *Candida* in oral biofilms [129]. 

Xerostomia is another treatment toxicity commonly observed in patients undergoing radiation therapy. Irradiated tissues have reduced blood supply, poor wound healing, and a subdued immune response. These changes result in the proliferation of anaerobic and microaerophilic bacteria in the supra and subgingival biofilms. Furthermore, saliva is a significant contributor to the stability of the oral microenvironment. For this reason, the loss of salivary production characteristic of xerostomia fosters imbalances in the oral microbiome. Specifically, patients who receive radiation therapy present with an oral cavity that contains a greater abundance of *Lactobacillus* species. Such bacteria are associated with the formation of caries, a known complication of hyposalivation [159,160].

Colitis is a common complication of checkpoint inhibitors. Such colitis is uncomfortable to patients and causes significant morbidity secondary to nutritional losses leading to treatment failure. Cornerstones in treating immune checkpoint inhibitor-induced colitis include glucocorticoids and other forms of immunosuppression. Dubin et al. studied a population of melanoma patients undergoing treatment with checkpoint inhibitors. They found that patients with abundant intestinal *Bacteriodetes* phylum were less likely to suffer from immune checkpoint inhibitor-induced colitis [161]. No similar studies in HNSCC were seen however this theoretically may be extrapolated to HNSCC given the wide-reaching effects of checkpoint inhibitors and the gut microbiome.

Associations between microbiota and treatment toxicities seen in the course of are summarized in Table 4.

### 4.4. The Microbiome as a Therapeutic Instrument

The possibility of microbial-based malignancy treatments has been drawing attention for hundreds of years. As far back as the 1700s, there are recordings that specific infectious processes could exert a protective or therapeutic effect on select malignancies. This was not lost on William B. Coley, a surgeon active between 1891 and 1936, who used a bacterial vaccine derived from heat-inactivated *Streptococcus pyogenes* and *Serratia*
*marcescence* to treat inoperable sarcoma with a cure rate of better than 10% [162]. 

Bacteriotherapy, therapy that utilizes bacteria strains, peptides, bacteriocins, and toxins, is a unique approach to cancer treatment [163]. This is the category in which treatment for non-muscle invasive bladder cancers in which the Bacillus Calmette-Guerin (BCG), anti-tuberculoid, vaccine resided. This treatment has been approved for use by the FDA and has been approved since the 1970s [164,165]. The role of bacteriotherapy is currently being investigated in gastrointestinal cancers, where there are many prospective therapeutics, with each therapeutic showing promise in combatting multiple cancers. Of the bacteriotherapy models reviewed by Soleimanpour and colleagues [163] only Nisin A and Exotoxin A show promise in head and neck cancers [163]. 

Nisin A is a peptide bacteriocin produced by *Lactococcus lactis* to kill other Gram-positive competitors in its environment. It has been widely used in food preservatives and is known to modulate the human immune system through both the innate and adaptive immune systems by acting on neutrophils and T cells [166]. Joo and colleagues demonstrated that nisin could induce preferential apoptosis, cell cycle arrest, and reduced cell proliferation in HNSCC cells, compared with primary keratinocytes in vitro and in vivo [167]. A study by Kamarajan et al. demonstrated prolonged survival of mice with squamous cell cancers of the head and neck who were treated with nisin versus controls [168]. 

Produced by *Pseudomonas aeruginosa*, Exotoxin A is an elongation factor 2 (EF-2) inhibitor that functions by ADP-ribosylation of EF-2, leading to inhibited protein synthesis and ultimately apoptosis of the affected cell [163]. A study by Thomas et al. demonstrated response in HNSCC cell lines treated with several low intratumoral doses of epidermal growth factor receptor ligand-transforming growth factor β (TGF-β) fused to *Pseudomonas* exotoxin in vivo and in vitro. The treatment was tolerated well in a mouse model [169]. 

Routy et al. studied the benefits of fecal microbial transplant (FMT) for patients with renal cell cancer, urothelial cancers, or non-small cell lung cancers receiving immunotherapy. In this study, FMT occurred from patients who responded well to PDL-1 inhibitors to experimental arm mice induced with malignancy. A second arm examined FMT from non-responders to control mice also induced with malignancy. Mice in the experimental arm had better outcomes than their peers in the control groups. Stool from responders demonstrated high levels of *Akkermansia muciniphila*. When mice treated with PDL-1 inhibitor and FMT from non-responders received oral supplementation of *A.*
*muciniphila*, which was found to restore the efficacy of the PDL-1 blockade [149]. A subsequent study inspired by Routy’s work demonstrated fecal *A.*
*muciniphila* was associated with clinical benefit in patients with non-small cell lung cancers and kidney cancers treated with checkpoint inhibitors [170]. Similar findings have been demonstrated in melanoma mouse studies where supplementation of *Faecalibacterium* spp., *Bifidobacterium longum*, or *Collinsella aerofaciens* from immunotherapy responders led to improved outcome of treatment with immunotherapeutic agents over controls who did not receive supplementation. This was echoed by another mouse study in which *Bifidobacterium* was found to promote antitumor activity and facilitate anti–PD-L1 efficacy in melanoma mouse models [7]. Overall, these studies support the use of FMT to create change in the gut microbiota that can be associated with improved response to immunotherapy at distant sites of malignancy [148]. Currently, several phase 1 and 2 trials are investigating the efficacy of FMT and single strain microbe introduction in patients receiving immunotherapy. Some trials generally focus on “solid tumors,” [8] but only one trial was identified as studying outcomes in HNSCC (specifically, locoregionally advanced oropharyngeal squamous cell cancers) treated with chemoradiation therapy with or without FMT. No data has been released from this trial at this time [171]. Nonetheless, a recent review by Gavrielatou et al. listed the microbiome as a host-based factor that could be optimized to obtain a response for those receiving immunotherapy for the treatment of HNSCC [172].

Supplementing the microbiome with exogenous microbiota with probiotics or FMT is not without some inherent risks. There are numerous reports of septicemia related to probiotic treatment [173,174,175,176]. For this reason, some have suggested supplementation with prebiotics to help foster the existing bacteria in a patient’s gut that may contribute to an anticancer immune response. For example, increased intake of certain dietary fibers can increase the number of butyrate-producing bacteria in a patient’s gut associated with improved outcomes [177].

Others have suggested designer microbes and microbiota transplant (aside from those microorganisms found naturally) as the future of the microbiome in successful oncologic management [178]. Additional research has focused on mitigation strategies for adverse events encountered in treating head and neck cancers. For example, many studies have investigated the use of probiotics to prevent mucositis, with mostly favorable results [179,180,181]. Additionally, a case series reported by Wang and colleagues demonstrated successful treatment of refractory immune checkpoint inhibitor-induced colitis with FMT [182]. Given the promise of pre-clinical and clinical data in other cancers and the relative dearth of data specific to HNSCC, more research will be needed to guide the future microbial-based treatments of HNSCC.

## 5. Conclusions and Perspectives

The complex relationship between the microbiome and the development and treatment of HNSCC is a field of great interest. This research is rapidly evolving, and as data builds, there is hope that the release of more literature will lead to advancing guidelines that will allow HNSCC to be detected at earlier stages, prevented in those with known risk factors, and treated with less interruption from the most frequently encountered adverse events. New research also describes how the microbiome modulates response to anticancer therapies. The data compiled here suggests that considering the relationship between the host and microbiome in HNSCC may lead to essential screening tools and treatment options that would be valuable in improving clinical outcomes. In this manuscript, the authors have summarized the complex relationship between HNSCC and the microbiome by addressing the normal microbiome of the areas affected by HNSCC and theories broadly aiming to explain the microbiome as a modulator of malignancy. The authors have also reviewed suggested biomarkers to be used as screening tools for HNSCC, the effect of HNSCC treatment on the microbiome, the role of the microbiome in HNSCC outcomes and treatment toxicities, and the microbiome as a therapeutic agent. In doing so, the authors have revealed several areas for continued research, as much of the data reviewed was collected in the investigation of other malignancies which may or may not be extrapolated to be applied to HNSCC.

Furthermore, most of the data reviewed were collected from preclinical trials, and the little clinical data available in the literature was unfinished. The authors recommend the identification of distinctive flora that increase the risk of HNSCC development and microbiota unique to early stage HNSCC and OPMD as fields for continued research that may prevent HNSCC entirely or lead to earlier detection of primary or recurrent HNSCC, thus preventing late-stage HNSCC. Moreover, focusing on the microbiome as a treatment tool through probiotics, prebiotics, FMT, and direct supplementation could mitigate some of the treatment toxicities and improve outcomes as outlined. The future of oncology may include utilizing these techniques and others, paving the way to a unique form of medicine reliant upon the microbiota not yet seen in oncology. 

## Figures and Tables

**Figure 1 cancers-14-04116-f001:**
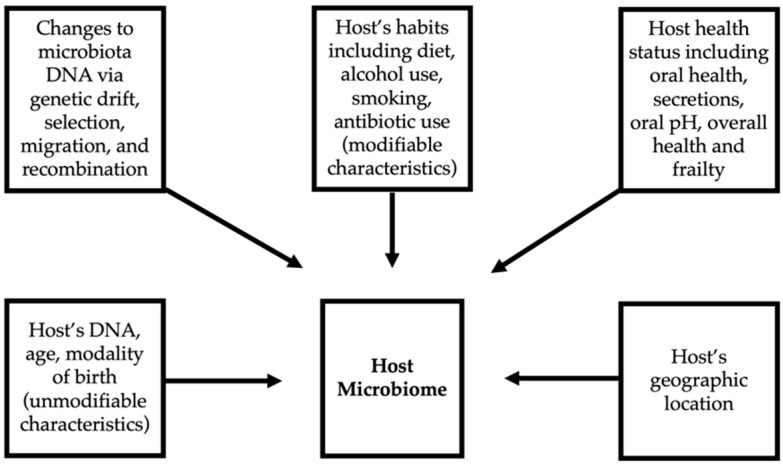
Factors influencing an individual’s microbiome.

**Figure 2 cancers-14-04116-f002:**
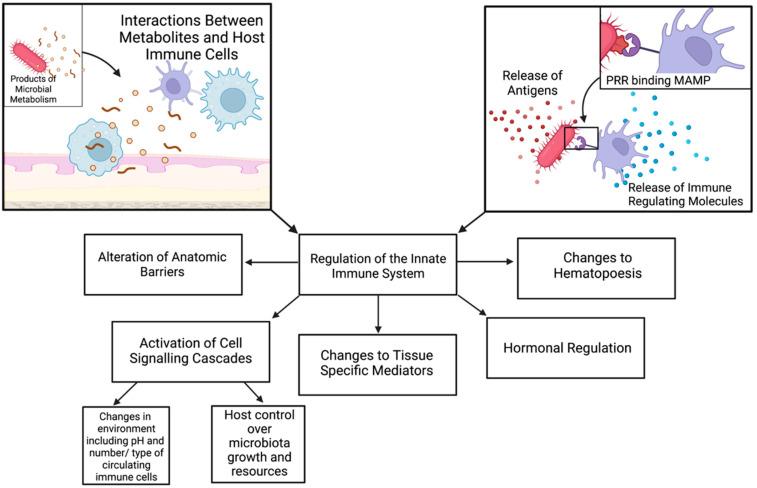
Methods through which the microbiome alters the innate immune system. Abbreviations: MAMP, microbe-associated molecular patterns; PRR, pathogen recognition receptor. Created with biorender.com (accessed on 18 August 2022).

**Figure 3 cancers-14-04116-f003:**
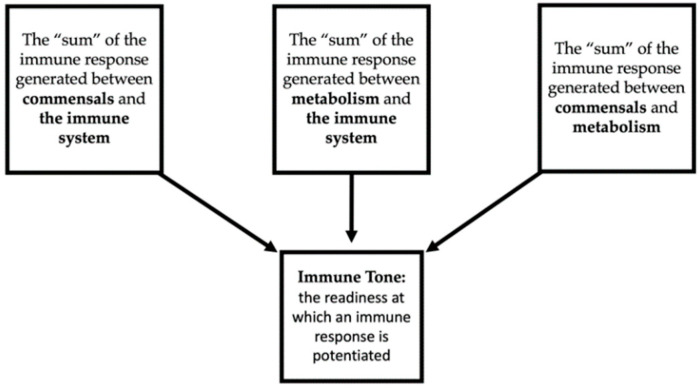
Factors influencing immune tone according to Chervonsky’s “Commenselocentric View of the Homeostatic Maintenance of Host Health”.

**Table 1 cancers-14-04116-t001:** A Summary of the Healthy, Premalignant and Malignant Microbiome of the Head and Neck.

Healthy			
**Flora**	**Technique**	**Notes**	**Source**
*Firmicutes*, *Bacteroidetes*, *Proteobacteria*, *Fusobacteria*, and *Actinobacteria*	16S rDNA V4 sequencing of Isohelix SK-2 swabs	*Firmicutes* and *Actinobacteria* reduced in malignant tissues	[21]
*Streptococcus*, *Haemophilus*, *Actinomyces*, and *Prevotella*	16S rRNA sequencing	Reportedly present in the healthy oral microbiome (generally)	[17,22]
*Neisseria*, *Haemophilus*, *Fusobacterium*, *Porphyromonas*	16S rRNA V2-V4 sequencing of oral swabs	Reportedly present in the healthy oral microbiome based on healthy controls in a gastric cancer study	[24]
*Streptococcus*	16S rRNA sequencing	Reportedly present in the healthy oral microbiome (generally)	[16]
*Actinomyces*	16S rDNA sequencing of paired normal and tumor resections	Concentration of *Parvimonas* positively correlated to T-stage	[48]
*Haemophilus*, *Corynebacterium*, *Paludibacter*, *Porphyromonas*, and *Capnocytophaga*	16S rRNA sequencing of oral rinse	Examiners were able to reliably predict the presence of oral cavitycancer and oropharyngeal cancers	[56]
*Rothia* and *Haemophilus*	16S rRNA sequencing of salivary samples	More prevalent in control patients than patients with HNSCC	[49]
**Premalignant**			
**Flora**	**Technique**	**Notes**	**Source**
*Cloacibacillus*, *Gemmiger*, *Oscillospira*, and *Roseburia*	16S rDNA V4 sequencing of saliva samples	Also present in patients with confirmed malignancy, but statistically decreased in healthy controls	[39]
*M*. *micronuciformis*	16S PCR V6-V9 sequencing of swabs	A partner of Fusobacterium in fostering the development of malignancy in the throat by changing the microenvironment and biofilm formation	[41]
*Prevotella melaninogenica*, *Porphyromonas*, and *Solobacterium*	16S rRNA V4 sequencing of salivary samples	Lower abundance of *Haemophilus*, *Corynebacterium*, *Cellulosimicrobium*, and *Campylobacter* in oral microbiota in comparison to healthy controls	[42]
**Malignant**			
**Flora**	**Technique**	**Notes**	**Source**
*Bacillus*, *Enterococcus*, *Parvimonas*, *Peptostreptococcus*, and *Slackia*	16S rDNA V4 sequencing of saliva samples	Increased in cases of malignancy when compared to oral potentially malignant disorders	[39]
*Parvimonas*	16S rDNA sequencing of paired normal and tumor resections	Concentration of *Parvimonas* positively correlated to T-stage	[48]
*Peptostreptococcus*, *Fusobacterium*, *Alloprevotella*, and *Capnocytophaga*	16S rRNA sequencing of salivary samples	More abundant when comparing the microbiome of cancer patients to the control patients	[49]
*Fusobacterium nucleatum*, *Pseudomonas aeruginosa*, and *Campylobacter*	16S rRNA V1-V3 s equencing of tissue samples	An overabundance of these microbiota were noted in tumor tissue when compared to healthy tissue	[50]
*Fusobacterium nucleatum*, *Capnocytophaga sputigena*, *Porphyromonas endodontalis*, and *Gemella haemolysans*	NGS of oral swabs	The relative concentration of *P. endodontalis*, *Gemella morbillorum*, and *G. haemolysans* related to increased depth of invaision	[51]
*Schlegelella* and *Methyloversatilis*	16S rRNA sequencing	Relative abundance of these organisms related to worse prognosis	[52]
*Prevotella*, Stomatobaculum, and *Bifidobacterium*	16S rRNA V1-V3 sequencing of salivary samples	With a relative loss of *Fusobacterium*	[53]
*Capnocytophaga gingivalis*, *Prevotella melaninogenica*, and *Streptococcus mitis*	NGS of salivary samples	Examiners were able to reliably predict the presence of malignancy based upon these organisms	[54]
*Oribacterium*	16S rRNA sequencing of oral rinse	Examiners were able to reliably predict the presence of oral cavity cancer and oropharyngeal cancers based on the presence of *Oribacterium*	[56]

Abbreviations: HNSCC, head and neck squamous cell cancer; NGS, next generation sequencing; PCR, polymerase chain reaction; T-stage, tumor stage; rDNA, recombinant deoxyribonucleic acid; rRNA, ribosomal ribonucleic acid.

**Table 2 cancers-14-04116-t002:** Effect of HNSCC Treatment on the Oral Microbiota.

Intervention	Associated Impact	Microbiota	Source
Surgery	Increased Levels	*Haemophilus*, *Neisseria*, *Aggregatibacter*, *Leptotrichia*	[127]
Radiation	Decreased Levels	Gram-negative obligate anaerobes	[129]
Radiation	Increased Levels	*Streptococcus mutans*	[129]
Radiation	Increased levels	*Actinomyces*, *Lactobacillus*, *Sreptococcus mutans*	[22]
Radiation	Decreased Levels	*Neisseria*, *Fusobacterium*, *Streptococcus sanguinis*	[22]
Chemoradiation	Increased Levels	Gut-associated taxa	[137]
Radiation	Increased Levels	*Candida*, enteric rods, *Staphylococci*	[131]
Chemotherapy	Decreased levels	Oral *Streptococci*	[16]
Chemotherapy	Increased levels	Oral Gram-negative anaerobes	[16]

**Table 3 cancers-14-04116-t003:** Effect of Microbiota on the Outcome of HNSCC Treatment.

Microbiota	Associated Impact	Outcome	Source
*Fusobacterium nucleatum*	Improved	Recurrence rate, overall survival, relapse free survival, metastasis free survival	[59]
*Porphyromonas gingivalis*	Increased	Chemoresistance	[146]
Butyrate producing microbes	Decreased	Radiotherapy effectiveness	[147]
*Akkermansia muciniphila*	Increased	Response to immune checkpoint inhibitors	[149]
*Peptostreptococcus*	Increased	Overall survival	[156]
*Bifidobacterium longum*, *Collinsella aerofaciens*, *Enterococcus faecium*	Increased	Response to immune checkpoint inhibitors	[154]
*Ruminococcaceae* family	Increased	Response to immune checkpoint inhibitors	[155]
Normal gut flora	Increased	Overall survival	[157]

**Table 4 cancers-14-04116-t004:** Association of Microbiota and Treatment Toxicities.

Microbiota	Associated Impact	Toxicity	Source
*Prevotella*, *Fusobacterium*, *Streptococcus*, *Megasphaera*, *Cardiobacterium*	Increased risk	Oral mucositis	[158]
Gram negative bacteria	Increased severity	Oral mucositis	[6]
*Enterobacteriaceae*, *Candida*	Increased severity	Oral mucositis	[129]
*Lactobacillus*	Increased risk	Dental Caries	[160]
*Bacteriodetes*	Decreased risk	Immune checkpoint inhibitor induced colitis	[161]

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
