# Peer review of "A Review of the Role of Oral Microbiome in the Development, Detection, and Management of Head and Neck Squamous Cell Cancers"

_cancers, 2022, doi:10.3390/cancers14174116_

Round 1

Reviewer 1 Report

This review is a good discussion based on previous reports on head and neck cancer and the bacterial flora, which has not received much light so far.

It is almost worthy of publication as is, but as a review related to head and neck cancer, it would be a good idea to summarize the results in a Table, especially the results mentioned regarding the oral microflora and its prognosis.

I also thought it would be good to have a section on comparisons with lung or esophageal cancer, which are contiguous organs with head and neck cancer.

Thank you for the opportunity to review such a valuable article.

Author Response

Reviewer 1,

Thank you for your kind words and thoughtful criticisms of our manuscript. We appreciate the time and effort that you have dedicated to providing your valuable feedback. We have incorporated changes to reflect most of the suggestions provided by all three reviewers. Here is a point-by-point response to your specific comments and concerns.

  • Comment 1: It is almost worthy of publication as is, but as a review related to head and neck cancer, it would be a good idea to summarize the results in a Table, especially the results mentioned regarding the oral microflora and its prognosis.
    • Response: Thank you for this suggestion, and we agree this would be an excellent way to help summarize the pertinent aspects of our review. Therefore, we have added four tables, each at the end of sections that they summarize, that provide information regarding the microbiota that comprise the healthy, malignant, and premalignant microbiomes of the head and neck and the impacts that different microbiota and aspects of head and neck cancer have on each other. These tables are at the end of sections 2.3, 4.1, 4.2, and 4.3.
  • Comment 2: I also thought it would be good to have a section on comparisons with lung or esophageal cancer, which are contiguous organs with head and neck cancer.
    • Response: Thank you for this suggestion, and this would indeed be an interesting aspect to consider. However, we believe this would be outside the scope of our paper and would be better addressed by a more comprehensive report that focused solely on those topics. We truly believe this idea has literary value but do not believe we would be able to provide an in-depth review of this topic within the current manuscript.

Sincerely, 

Kimberly Burcher and Colleagues

Reviewer 2 Report

This present review article by Burcher et al establishes the connection between microbiome and malignancy (HNSCC).  Also they address the key issues in treatment and its related toxicities in patient’s with HNSCC. I am in principle supportive of accepting this work for publication. However, I have few suggestions to improve the manuscript for publication. 

Major 

  1. I strongly encouraged authors to use table format to layout samples tested, techniques and main findings in clinical samples along with references.

  2.  For explaining section 3, the author can use cartoons to describe the cross communication of bacteria and immune system in HNSCC. 

  3.  Cartoons can be used whenever necessary. 

Minor 

  1. Ample of spelling mistake was detected through the text, mostly Auerobasidum should be corrected to Aureobasidium, Peptosteptococcus should be corrected to Peptostreptococcus, Bordatella should be corrected Bordetella, Serattia should be corrected to Serratia

  2. Spelling mistakes and grammatical errors occurs more frequently throughout the text. 

  3. Scientific names are always italicized, the genus is always capitalized, the species is never capitalized.

Author Response

Reviewer 2,

Thank you for your insightful thoughts on our manuscript. We appreciate the effort that you have dedicated to providing your valuable feedback. We have incorporated changes to reflect most of the suggestions provided by all three reviewers. Here is a point-by-point response to your specific comments and concerns.

Major 

Comment 1: I strongly encouraged authors to use table format to layout samples tested, techniques and main findings in clinical samples along with references.

  • You will find that we have constructed this table as recommended. It is listed as “Table 1” and appears on page 7. We have additionally added three other tables.

Comments 2 and 3: For explaining section 3, the author can use cartoons to describe the cross communication of bacteria and immune system in HNSCC. Cartoons can be used whenever necessary. 

  • We agree with this comment and have therefore added three figures. Figures 2 and 3 were specifically created to address your comment related to section 3.

Minor 

Comments 1-3: Ample of spelling mistake was detected through the text, mostly Auerobasidum should be corrected to Aureobasidium, Peptosteptococcus should be corrected to Peptostreptococcus, Bordatella should be correctedBordetella, Serattia should be corrected to Serratia. Spelling mistakes and grammatical errors occurs more frequently throughout the text. Scientific names are always italicized, the genus is always capitalized, the species is never capitalized.

  • Thank you for pointing this out. We agree with these comments and have corrected the names' formatting, spelling, and capitalization.

We are very grateful for your time and would like to thank you for working with us to improve our manuscript.

Sincerely, 

Kimberly Burcher and Colleagues

Reviewer 3 Report

The authors performed a very comprehensive review of the literature on the role of the oral microbiome on the development, detection, and management of head and neck squamous cell carcinomas. This reviewer thought this review was very well written, comprehensive, and of significant value and interest to the scientific community that is interested in the impact of the microbiome within the context of head and neck cancers. I recommend this manuscript for publication pending very minor revisions.

Minor Issues:

1. Lines 17-18: "they had a neck". I think this is supposed to say head and neck.

2. Several places instead of OPMD, it is written as OMPD.

3. Line 283: Missing beta symbol for beta-catenin

4. line 432: "pharmaceuticals.34". I think this was supposed to be a reference but isn't in proper format.

5. line 781: Missing beta symbol for TGF-beta

Author Response

Reviewer 3,

We sincerely appreciate your kind review and thank you for the time and effort you have dedicated to improving our manuscript. We have incorporated changes to reflect most of the suggestions provided by all three reviewers. We have addressed all five of the grammar mistakes, spelling mistakes, and typos raised by your review. We are very grateful for your time.

Sincerely, 

Kimberly Burcher and Colleagues

Round 2

Reviewer 2 Report

This revised version substantially improves the quality of their research work. The Authors tried their best to correct all the mistakes and provided a revised version of their review. Therefore, I accept this work for publication.